# Polymer-Based Self-Assembled Drug Delivery Systems for Glaucoma Treatment: Design Strategies and Recent Advances

**DOI:** 10.3390/polym15224466

**Published:** 2023-11-20

**Authors:** Hao Sun, Guangtong Wang, Qingying Feng, Shaoqin Liu

**Affiliations:** 1School of Chemistry and Chemical Engineering, Harbin Institute of Technology, Harbin 150080, China; 19b925120@stu.hit.edu.cn; 2Zhengzhou Research Institute, Harbin Institute of Technology, Zhengzhou 450000, China; 3School of Medicine and Health, Harbin Institute of Technology, Harbin 150080, China; 4School of Life Science and Technology, Harbin Institute of Technology, Harbin 150080, China; fqy9502@mail.ru

**Keywords:** self-assembled nanocarrier, drug delivery, glaucoma, polymers, intraocular pressure

## Abstract

Glaucoma has become the world’s leading cause of irreversible blindness, and one of its main characteristics is high intraocular pressure. Currently, the non-surgical drug treatment scheme to reduce intraocular pressure is a priority method for glaucoma treatment. However, the complex and special structure of the eye poses significant challenges to the treatment effect and safety adherence of this drug treatment approach. To address these challenges, the application of polymer-based self-assembled drug delivery systems in glaucoma treatment has emerged. This review focuses on the utilization of polymer-based self-assembled structures or materials as important functional and intelligent carriers for drug delivery in glaucoma treatment. Various drug delivery systems, such as eye drops, hydrogels, and contact lenses, are discussed. Additionally, the review primarily summarizes the design strategies and methods used to enhance the treatment effect and safety compliance of these polymer-based drug delivery systems. Finally, the discussion delves into the new challenges and prospects of employing polymer-based self-assembled drug delivery systems for the treatment of glaucoma.

## 1. Introduction

Glaucoma is a degenerative disease of the optic nerve characterized by changes in the optic nerve and damage to the visual field, caused by various underlying factors [1]. It has now emerged as the leading cause of irreversible blindness worldwide [2,3,4]. The term “irreversible blindness” indicates that once glaucoma damages or blinds a patient’s visual function, current technology cannot fully restore their vision to its previous level. Globally, glaucoma primarily affects individuals over the age of 40, encompassing approximately 3.54% of the total population. Alarmingly, it is projected that the number of individuals afflicted by glaucomatous blindness will reach 118.8 million by 2040 [3,5,6]. This condition not only significantly impacts an individual’s quality of life but also imposes substantial personal and socio-economic burdens [7]. As a result, glaucoma is poised to become a critical public health issue in the coming decades, demanding heightened attention and awareness [8].

The biological basis of glaucoma has not been completely figured out, and the factors contributing to its occurrence and development are not fully understood, thereby limiting clinical treatment options [5]. In recent years, long-term clinical trials completed by researchers have provided convincing evidence linking intraocular pressure (IOP) to retinal ganglion cell death, indicating that reducing IOP can effectively prevent the progression of early and late glaucoma [9,10,11]. Consequently, researchers consider reducing IOP as the primary and most effective method for glaucoma treatment [6]. The main goal of glaucoma treatment is to delay the progression of the disease and preserve the quality of life of patients. Clinicians believe that achieving the target IOP level can effectively slow down the advancement of glaucoma and prevent further functional impairment. The determination of the target IOP level for a specific eye depends on factors such as the patient’s IOP, the extent of fundus damage, and visual field impairment [9,12]. It is essential to achieve the target IOP level with the minimum number of drugs, minimal adverse reactions, and optimal patient compliance. Glaucoma treatment mainly involves the use of drugs or surgery to reduce high IOP in patients, with non-surgical drug treatment being the primary approach employed at present [2,13,14,15].

The eye is one of the most complex organs in the body. The unique structure of the eye forms the anatomical and physiological barrier of the eye. This barrier impedes drug penetration, resulting in low drug bioavailability and serving as a major obstacle in glaucoma drug treatment [16,17,18]. In the non-surgical drug treatment scheme to reduce intraocular pressure, that is, the ocular surface drug administration regimen, traditional eye drops are widely used [10]. Ocular surface administration primarily aims to cross the eye’s barrier and reach the targeted tissue through the corneal route to exert its therapeutic effects (see Figure 1). Studies have revealed that most eye drops are washed away by tears within 15~30 s, leading to reduced drug residence time and a bioavailability of less than 5%, consequently lowering efficacy [16,19]. Furthermore, even if the eye drop breaks through the tear film barrier, it encounters the challenge of crossing the cornea, which is composed of the epithelial layer, Bowman’s layer, cornea stroma, Descemet’s membrane, and endothelial cells. The closely connected cells in the epithelial layer make it difficult for drugs to pass through [20]. The high lipid content in the cornea makes it even more challenging for hydrophilic drug molecules to penetrate. The matrix layer, being hydrophilic, restricts the passage of hydrophobic drug molecules. Adherence is crucial for the effective treatment of glaucoma. Studies have shown that many glaucoma patients struggle to maintain high compliance, and patients with compliance of 21% demonstrated progressive visual field defects [16]. Poor drug compliance in glaucoma often results from reduced frequency of eye drop usage due to patient forgetfulness, difficulties with self-administration, and significant side effects. Therefore, an effective drug delivery system plays a crucial role in improving drug compliance. [14,21,22,23].

In recent years, the rapid development of drug encapsulating systems with polymers as carriers has been fueled by the significance of polymers as essential structural, functional, and intelligent materials, coupled with their excellent biocompatibility. A polymer drug delivery system is defined as a preparation or device that efficiently introduces therapeutic substances into a specific targeted location [24]. Numerous studies have demonstrated that polymer drug delivery systems can achieve continuous drug delivery to the eye, effectively enhancing the bioavailability and compliance of drugs [25,26,27,28,29].

This review focuses on the application of polymer-based self-assembled drug delivery systems in glaucoma treatment. It aims to enhance the effective utilization and adherence of polymer drug delivery systems to improve treatment outcomes, primarily by describing and summarizing the strategies and methods used in the design of such systems. Additionally, the review discusses the potential new opportunities for utilizing polymer drug delivery systems in glaucoma treatment.

## 2. Polymer-Based Delivery Systems for Glaucoma Treatment

The primary objective of glaucoma drug treatment is to reduce intraocular pressure, requiring most patients to undergo long-term medication for its control. However, the treatment’s effectiveness is often hindered by the low bioavailability and poor compliance of drugs. Thus, the development of a drug delivery system that is capable of sustained release becomes crucial [10,30]. In clinical settings, glaucoma drugs can be categorized based on their mode of action. Some drugs, such as α-adrenergic agonists, β-receptor blockers, and carbonic anhydrase inhibitors, work by reducing aqueous humor production. Others, such as prostaglandins, rho kinase inhibitors, nitric oxides, and mitotic or cholinergic drugs, facilitate fluid discharge from the eye. The polymer self-assembling drug delivery system represents an advanced and versatile drug delivery technology. This system offers precise control over drug release rates and timing, making it a valuable tool for achieving highly controllable drug release properties. Moreover, polymers play a pivotal role in shielding drugs from physical and chemical degradation, enhancing drug stability, and preserving their potency. One of the key advantages of polymer self-assembling drug delivery systems is their ability to encapsulate drugs within polymeric nanoparticles. This encapsulation can significantly enhance drug bioavailability. Furthermore, these systems can be meticulously engineered to release drugs under specific biological conditions. By doing so, they enable more precise drug delivery to target tissues, ultimately improving the precision of therapeutic targeting. An additional benefit of polymer-based self-assembling drug delivery systems is their potential to reduce the frequency of medication doses. These systems achieve this by prolonging the release of drugs within the body, ultimately enhancing patient convenience and treatment adherence.

Polymeric drug delivery systems offer a solution to overcome the multiple barriers presented by the eye, enabling enhanced drug attachment, permeability, and sustained release on the cornea. To achieve this, several key strategies can be employed in the design of polymeric drug delivery systems: (1) Choose polymeric materials known for their excellent biocompatibility and biodegradability. This not only enhances the safety profile of polymeric carriers but also ensures minimal adverse effects. (2) Enhance the affinity of the polymer carrier to the cornea through surface modifications. This step helps improve the interaction between the carrier and the corneal tissue. (3) Boost the drug’s corneal permeability by incorporating penetration enhancers into the formulation. These enhancers facilitate the drug’s ability to traverse the corneal barriers effectively. (4) Utilize appropriate drug encapsulation technologies to ensure efficient drug loading, thereby achieving controlled release and prolonged therapeutic effects. (5) Leverage nanotechnology to create drug delivery nanoparticles with high surface area and permeability, enhancing the drug’s penetration into the cornea. (6) Design environmentally responsive polymer carriers capable of achieving targeted drug release in response to various ocular environmental conditions. These strategies collectively aim to prolong drug residence time on the cornea, improve permeability, and mitigate systemic side effects, ultimately enhancing the effectiveness of ocular drug delivery.

The use of polymer-based self-assembled drug delivery systems allows for the effective and long-term delivery of one or more drugs to treat glaucoma [31]. Herein we focus on polymer-based self-assembled drug delivery systems, especially non-surgical and non-implanted systems positioned on or near the ocular surface, including Eye drops [32,33], Hydrogels [34,35], and Contact lenses [36,37], all incorporating polymers. These drug-encapsulating systems function as effective drug delivery mechanisms, ensuring sustained drug release. The design strategies and methods employed for these systems hold immense significance (see Figure 2).

### 2.1. Eye Drops

Eye drop administration is the most convenient method to deliver ocular surface drugs for treating glaucoma with ocular hypertension. However, blinking, tear flushing, and the complex anatomical structure of the tear film and cornea reduce the bioavailability of drugs, leading to poor drug compliance [38]. To address these issues, the design strategy for eye drops should focus on increasing the residence time of drugs in the cornea, enhancing bioavailability, improving therapeutic effects, and ensuring good histocompatibility to enhance drug compliance [39]. During the process of physiological blinking, ocular shear force and tear wash are generated, which enables polymers or pseudoplastic fluids with appropriate viscosity to maintain higher drug concentrations on the ocular surface. Mucins present on the ocular surface attach to the tear film surface and carry a negative charge. This makes it more likely that positively charged eye drops and drug carriers will remain on the ocular surface. The design strategy for eye drops should involve selecting an appropriate polymer with good biocompatibility, adjusting the polymer carrier’s structure to endow it with appropriate viscosity and positive charge, and ensuring efficient drug loading capabilities. These modifications are conducive to increasing drug retention time in the cornea, thus improving bioavailability. Moreover, the sustained release of drugs further enhances medication compliance and ultimately improves the therapeutic effect [40].

Increasing the viscosity of eye drops is a straightforward approach to extend the retention time of drugs on the ocular surface, thereby enhancing their bioavailability [19]. At present, one of the methods to achieve higher viscosity is by using polymer thickeners, such as hyaluronic acid, polyvinyl alcohol, cyclodextrin, and others [41]. For example, Alviset et al. employed sodium hyaluronate as a thickener and polysorbate 80 as a surfactant to self-assemble the system into micelles to load the drug, enhancing the performance of the eye drops, leading to a reduced dose of travoprost in the eye drops while improving bioavailability in rabbits and minimizing side effects [42]. Furthermore, nano or micro-structures formed by polymer self-assembly can not only increase the viscosity of the solution but also be applied as a carrier to controllably deliver the drugs. Battistini et al. utilized a complex of sodium hyaluronate and timolol maleate (TM) to reduce intraocular pressure in rabbits for 10 h. Similarly, Shahab et al. developed dorzolamide-loaded self-assembled polycaprolactone nanoparticles, which were coated by chitosan (DRZ-CS-PCL-NPs), resulting in adhesion strength 3.7 times stronger than that of the control group, facilitating better penetration into goat cornea [43]. Cyclodextrin, a cyclic oligosaccharide polymer, can also be used as a thickener. Researchers have shown particular interest in their hydrophobic and hydrophilic structures, which can efficiently load drugs through host-guest interaction allowing hydrophobic drugs to be fixed in the core while the hydrophilic outer layer can traverse the tear film to reach the cornea, thereby improving the bioavailability of eye drops [19].

Enhancing adhesion to the mucin layer is an effective means to increase the retention time of drugs in the cornea. Researchers have predominantly chosen polymer chitosan with cationic groups due to the negative electrical characteristics of the mucin layer [44]. Mohan et al. developed chitosan micelles capable of interacting with mucins on the corneal surface by polymer self-assembly, loaded with the IOP lowering drug brinzolamide, with a micelle particle diameter of 74.32 ± 1.46 nm, resulting in improved corneal permeability [45]. Nguyen et al. utilized the amination level effect of chitosan to enhance the formed polymeric self-assembled colloidal coating. This modification facilitated the swift passage of pilocarpine-loaded ceria nanocapsules through the corneal barrier and enabled precise, controlled drug release. In a rabbit model of acute glaucoma, a single local instillation effectively reduced intraocular pressure [46]. Inspired by lollipops, Wang et al. developed sodium alginate chitosan-based polymer self-assembly multilayer hydrogel spheres coated with zinc oxide-modified biochar by electrostatic interaction. The multilayer structure was applied as a programmable drug delivery system for TM and levofloxacin(LVFX), achieving increased drug loading and sustained stable release while enhancing adhesion (see Figure 3) [47]. In recent years, polyamide amine dendrimers have emerged as a promising choice for ophthalmic drug carriers due to their radiating molecular structure and highly branched configuration. A substantial number of groups on their surface can be designed as amino groups [31]. Due to the highly branched structure, dendrimers have a relatively large surface area and more binding sites, which can interact with molecules in the mucin layer of the ocular surface. This highly branched structure enables dendrimers to interact more effectively with the mucin layer of the ocular surface, thereby improving drug adhesion and permeability, ultimately facilitating efficient drug delivery in the eye. Bravo-Osuna et al. utilized the high-affinity interaction between cationic carbosilane dendrimer (G3-C) and ocular surface mucin layer to design an eye drop with low-concentration polymer self-assembled carrier, successfully delivering acetazolamide (ACZ). In the application of IOP lowering in rabbits, the onset time was only 1 h, with up to 7 h of sustained hypotensive effect, and exhibited good biocompatibility [48]. Yang et al. reported the use of a drug coupling method to prepare a drug based on polyamidoamine dendrimer G3 timolol analog, which effectively penetrated the cornea to reduce intraocular pressure. Approximately 8% of the drug penetrated the cornea within 4 h, showing high water solubility and non-toxicity [49].

The particle size of the polymer carrier in eye drops plays a crucial role in drug delivery efficiency. The scope of nanomedicine encompasses dimensions ranging from 1 to 1000 nm [50]. This dimension range is crucial because drugs are loaded into nanoparticles at the nanoscale, offering several advantages. These advantages include improved drug pharmacokinetics, and pharmacodynamics, reduced non-specific toxicity, lower immuno-genicity, and enhanced biocompatibility. Ultimately, these benefits contribute to the overall therapeutic efficacy of drugs [51,52]. Nanoscale drug delivery systems hold significant potential in the treatment of glaucoma [53]. Ali et al. developed a chitosan-based drug delivery system with nanoparticles measuring approximately 243 nm in size. This innovation effectively extended the time the drug remained in the cornea, resulting in a reduction of intraocular pressure in glaucoma patients. However, it is worth noting that after in vivo metabolism and distribution studies, the drug delivery system was swiftly cleared from the corneal region just six hours following intraocular administration. It then proceeded to traverse the nasolacrimal drainage system, ultimately entering the systemic circulation [54]. Additionally, another noteworthy development involves drug carrier nanoparticles with particle sizes ranging from 108.0 ± 2.4 to 257.2 ± 18.6 nm. These nanoparticles exhibited remarkable efficiency in penetrating the cornea, a crucial aspect of successful drug delivery in ophthalmology. The smaller size of these nanoparticles not only facilitates corneal penetration but also ensures the overall stability of the drug delivery system. Importantly, long-term stability was confirmed as the system remained unchanged even after three months of storage under various conditions [55]. Wang et al. conducted a study to assess the delivery efficiency and effectiveness of a drug-loading system loaded with brimonidine tartrate (BT) and TM using self-assembled nano-in-nano dendrimer hydrogel particles with different particle sizes as carriers. The results indicated that the drug loading system with nanoscale particles (200 nm) exhibited significant improvements in cell compatibility, degradation, drug release kinetics, and corneal permeability compared to the micron-scale (3 μm and 9 μm) drug loading systems. Furthermore, the drug loading system with nanoscale particles (200 nm) showed a 17-fold increase in the corneal permeability of drugs and achieved zero-order release kinetics for anti-glaucoma drugs, surpassing the performance of ordinary eye drops [56].

### 2.2. Hydrogels

Hydrogels, which are formed by the crosslink and self-assembly of polymers or low molecular weight molecules, with hydrophilic functional groups that can bind a large number of water molecules, making them ideal drug carriers with flexible characteristics similar to living biological tissues. In the treatment of glaucoma, reducing the frequency of medication to achieve therapeutic effects is a common approach to improve patient compliance. Due to their unique structure and properties, eye hydrogels have the potential to replace conventional eye drops, offering promising benefits [35,57]. In the design of hydrogels for glaucoma treatment, researchers have focused on developing ocular environmentally responsive gels that respond to factors such as temperature, pH, and ions. Key parameters, including in situ gel-forming ability, drug encapsulation and release, ocular biocompatibility, and biodegradation, are carefully considered when selecting suitable polymer hydrogel systems. These chosen systems can construct ocular hydrogel drug-loading systems with environmental responsiveness, in situ gel-forming ability, high drug loading, effective drug delivery, long-term drug release, and good biocompatibility, ultimately enhancing the bioavailability of drugs and patient compliance [25,31,58,59].

Ophthalmic thermosensitive hydrogels are widely used, with physiological temperature serving as a response factor for in-situ gel formation [60]. Typically, ocular thermosensitive hydrogels are composed of biocompatible polymers. Upon contact with the eye, these hydrogels swiftly transition into a gel state, extending their retention time within the eye and ensuring long-lasting adhesion. Khallaf et al. developed a thermosensitive in-situ gel using poloxamer 407 and adhesive hydroxypropyl methylcellulose, loaded with a fasudil complex with phospholipid. In a solution, surfactant molecules undergo self-assembly to create micellar structures once the polyoxyethylene poloxopropylene block copolymer Poloxamer 407 reaches the Critical Micelle Concentration value. This formulation improved the bioavailability of the drug and significantly reduced intraocular pressure in a rabbit glaucoma model [54]. Fedorchak et al. constructed a hydrogel drug loading platform using a thermoresponsive n-isopropylacrylamide and poly acid microspheres, leading to continuous brimonidine tartrate release for 28 days [61]. Cuggino et al. prepared nanogels based on n-isopropylacrylamide and acrylic acid-crosslinked nanoparticles by precipitation/dispersion polymerization method and polymer self-assembled technology, prepared nanogels through ionic interaction to load TM, which effectively reduced intraocular pressure and maintained it for 48 h in a rabbit glaucoma model [62]. Ozturk et al. used thermosensitive amphiphilic poly (ε-caprolactone)-poly(N-vinylcaprolactam-co-N -vinylpyrrolidone) graft copolymers to construct a drug-loading platform for the combined administration of two drugs. They successfully delivered indomethacin and dorzolamide to effectively treat glaucoma, demonstrating that the polymer matrix micelle system effectively adhered to the ocular surface and continuously released drugs. This outcome can be attributed to the unique water-incorporated macromolecular network structure of the temperature-sensitive amphiphilic polymer and the numerous hydrogen bonding interactions involved [63]. In the context of eye drops, it has been previously discussed that chitosan is a cationic polymer that can increase adhesion to the cornea. Pakzad et al. used a mixture of sodium bicarbonate and β-glycerophosphate as a gelling agent to prepare a chitosan-quaternized transparent thermosensitive in situ hydrogel. This hydrogel was based on the self-assembly of quaternary ammonium chitosan biomacromolecules loaded with TM, achieving stable drug release for more than a week(see Figure 4a) [64]. Badran et al. developed a thermosensitive in situ gel drug-loading platform with chitosan-loaded liposomes encapsulating metoprolol particles. Compared with the drug loading platform without chitosan, the formulation exhibited 4.4 times improved corneal permeability and significantly reduced intraocular pressure in rabbits [65].

Ocular pH-responsive hydrogels utilize the ocular pH as a key factor for in situ gel formation. These pH-sensitive hydrogels exhibit responsiveness to pH due to the presence of acidic or basic groups within the polymer network that can undergo ionization. For example, a common polymer like Carbopol^®^ 934P contains carboxyl groups that play a pivotal role in the polymer’s self-loading process. Allam et al. successfully prepared a pH-responsive in situ gel by loading betaxolol with Carbopol^®^ 934P and hydroxyethyl cellulose. This gel exhibited high viscosity and adhesion, leading to improved drug retention and bioavailability in the eye. In glaucoma rabbit eyes, the relative bioavailability of the drug with this gel compared to free betalol eye drops was as high as 254.7% [67]. Vijaya et al. developed a TM nanoemulsion with a pH-responsive in situ gel by incorporating polymer Carbopol^®^ 934P. The preparation remained in a liquid state at pH 4 and rapidly formed a gel when the pH increased to 7.4 (tear pH). This pH-triggered in situ gelation mechanism provided continuous drug release over a 24 h period [68].

Ion-responsive hydrogels utilize ions in the ocular environment as a key factor for in-situ gel formation. When compared with thermosensitive and pH-sensitive in situ gel delivery systems, ion-sensitive in situ gels typically require lower polymer concentrations to form a gel due to the involvement of ions. This reduction in polymer concentration helps minimize the amount of polymer needed for ocular applications, lowering the potential risk of adverse reactions. Additionally, ion-responsive hydrogels do not need to be designed with a significant pH difference from the natural pH of tears, which is 7.4. This reduces eye irritation and alleviates patient discomfort. Currently, the widely utilized ion-responsive polymer is gellan gum. Gellan gum, a linear polymer, undergoes a significant transformation in the presence of an electrolyte. It fosters the creation of an abundance of hydrogen bonds within the system, facilitating the cross-linking of gellan gum chains and resulting in the formation of gels. Xu et al. used 0.45% gellan gum as the gel matrix to prepare an ion-sensitive in situ gel for the treatment of glaucoma with brimonidine tartrate. Studies have demonstrated that the higher bioavailability of this gel in rabbit eyes is attributed to the retention effect of the ion-sensitive in situ gel mixed with tear fluid [69]. Shukr et al. developed a new type of ocular ion-responsive in situ mucoliposome gel using gellan gum and Carbopol^®^ 934 as matrix gel to prolong and enhance the therapeutic effect of travoprost [70].

To enhance the bioavailability and compliance of drugs, researchers have developed a dual environmental-responsive ocular hydrogel drug loading platform through a self-assembling cross-linking strategy. Rawat et al. created a temperature/ion-responsive in situ gel for the adrenergic antagonist nebivolol (NEB). The gel employed poloxamers (poloxamer-407 and poloxamer-188) as a thermoresponsive component and kappa-carrageenan as an ion-sensitive component. The optimized dual-responsive in situ gel exhibited desired flow characteristics at room temperature and rapidly underwent a sol-gel transition in the presence of simulated tear fluid (STF) at physiological temperature. The dual-responsive in situ gel achieved a sustained-release rate of 86% over 24 h, and it was well-tolerated in the eye, effectively treating glaucoma (see Figure 4b) [66]. Patel et al. developed a pH-sensitive in situ gel loaded with TM and gellan gum and Carbopol as matrix gel for the treatment of glaucoma. The preparation was carried out in a safe, efficient, and stable container closure device, improving the storage stability of the gel [71].

Furthermore, researchers have explored various methods to promote the application of eye gel. Tambe et al. successfully developed an in situ gel loaded with TM and dorzolamide hydrochloride using hot melt extrusion technology for glaucoma treatment. This approach solved the problem of mass production of in situ gel, extended the retention time, and improved the bioavailability [72]. Andreadis et al. prepared an in situ gel nanofiber membrane containing polyvinyl alcohol and poloxamer 407 using electrospinning technology for intraocular delivery of TM, effectively reducing intraocular pressure for 24 h [73]. Kim et al. employed an ion permeation strategy to successfully deliver latanprost in the poly(lactic-co-glycolic acid) gel system to the eye, demonstrating drug efficacy in vivo for more than 7 days [74].

### 2.3. Contact Lenses

Contact lenses offer the advantage of easy wear and direct contact with the tear film on the ocular surface. With drug loading capabilities, they serve as an alternative to traditional eye drop administration, making them an attractive drug delivery system for glaucoma treatment requiring long-term effective drug delivery to maintain normal intraocular pressure. Studies have reported that contact lens drug delivery systems can extend drug release for several days or even months [35,37,75]. When designing contact lens drug loading systems, the drug addition and loading method influence the transparency, swelling, and adhesion of the lenses. Common drug loading methods include soaking drug loading, molecular imprinting technique, and direct loading of drug colloidal particles. Choosing an appropriate drug loading method can improve the long-term effective drug release, bioavailability, and compliance [76].

The immersion drug loading method for contact lenses is a feasible and simple approach. For instance, Costa et al. successfully impregnated two anti-glaucoma drugs (acetazolamide and TM) into commercial silicon-based polymer hydrogel contact lenses (Balafilcon A) using intermittent supercritical solvent impregnation, obtaining a favorable drug release performance [77]. Yu et al. also explored drug immersion and release with five other commercial contact lenses, confirming the feasibility of the drug immersion method and demonstrating that polymer modification with ocular mucin layer adhesion can improve the contact lens surface performance [76]. Hosseini et al. reported loading TM into polymer nanoparticles by coupling chitosan with lauric acid and sodium alginate, which were subsequently added to the precursor for contact lens preparation. This resulted in silica gel contact lenses loaded with drugs. The modification of chitosan with lauric acid was crucial to enhance its hydrophobic properties, which proved beneficial for achieving compatibility between the nanostructure and the silicone matrix. The self-assembly and gelation processes were facilitated by harnessing the interaction between the positive charge of chitosan and the negative charge on the surface of sodium alginate, ultimately leading to a more stable polymer nanoparticle structure. Subsequent treatment with oxygen plasma irradiation and albumin immersion rendered the lens surface more hydrophilic and conducive to comfortable wear. The positively charged carrier system prolonged drug retention in the cornea and improved bioavailability, with a release duration of up to 6 days. Directly loading drug colloidal particles effectively enhanced drug release performance [78]. Dang et al. employed Precirol ATO 5 as the lipid phase, with soy lecithin and Pluronic^®^ F127 serving as surfactants. Additionally, polyethylene glycol monostearate was incorporated into the formulation. They utilized the solvent evaporation method to induce the self-assembly of the components, resulting in the preparation of PEGylated solid lipid nanoparticles loaded with latanoprost. This formulation exhibited the capability to achieve sustained release of latanoprost, making it a promising candidate for the treatment of glaucoma [78]. To enhance the performance of contact lenses, Anirudhan et al. embedded carboxymethyl chitosan hydroxyethyl methacrylate polyacrylamide TM molecularly imprinted copolymer onto the poly HEMA matrix, creating a therapeutic contact lens for sustainable TM administration. The drug can be reloaded through contact lens immersion, making it reusable [79]. Lee et al. added T)-loaded thermosensitive poly(N-isopropylacrylamide) nanogel (30~50 nm) to the contact lens via soaking or centrifugation plus soaking. The stability of the self-assembled nanogels was ensured by hydrogen bonding interaction. The lens maintained light transmission and oxygen permeability while exhibiting temperature-triggered sustained drug release at 35 °C, providing environmental responsiveness [80]. Kim et al. loaded TM into a biocompatible polymer polyvinyl alcohol to construct highly integrated intelligent contact lenses for diagnosis and treatment with electronic sensors. Experiments in glaucoma model rabbits confirmed the feasibility of using intelligent contact lenses for IOP monitoring and control, suggesting this as a future development trend for contact lenses in glaucoma treatment (see Figure 5) [81].

## 3. Summary and Outlook

Glaucoma causes irreversible blindness and significantly impacts patients’ health and quality of life, resulting in a considerable economic burden on both individuals and society. Currently, the main treatment method to alleviate glaucoma’s progression involves drug therapy to reduce high IOP. However, due to the multiple barriers of the eyes, achieving effective treatment and patient compliance with drug therapy has become a major challenge. Urgent action is needed to develop more efficient and patient-friendly solutions. This review introduces the application of polymer drug delivery systems in glaucoma treatment, including eye drops, ophthalmic hydrogels, and contact lenses, and discusses design strategies and methods to improve their therapeutic effectiveness and compliance.

Eye drops are a more widely accepted treatment option for the public, but they currently suffer from issues such as low drug utilization and poor compliance with frequent administration, leading to suboptimal therapeutic outcomes for glaucoma. The review proposes exploring intelligent polymer carriers with stronger targeted delivery capabilities and controlled and effective drug release to address this issue. This will require researchers to better understand the biological anatomy of the eye, identify targeted drug tissues to reduce glaucoma-induced high IOP, consider specific environmental conditions (e.g., temperature, pH, ions) and protein and gene expressions at the targeted site, and select the appropriate polymer matrix. Regulating the structure and functional groups will enable targeted drug delivery.

Hydrogels and contact lenses loaded with intraocular-pressure-lowering drugs have emerged as the primary alternatives to traditional eye drops in glaucoma treatment. The review describes the delivery scheme of ophthalmic hypotensive drugs using hydrogels in response to the ocular environment. Compared to hydrogels with single ocular environmental response factors, hydrogels with multiple responses offer more advantages in improving drug bioavailability and compliance. However, this undoubtedly increases the complexity of hydrogel materials. Therefore, it is necessary to optimize the materials based on the relationship between the structure and properties of responsive polymers, allowing less content of the drug hydrogel carrier to achieve better response performance. Additionally, appropriate preparation methods must be developed to ensure hydrogel stability and biological safety while enhancing the therapeutic effect for glaucoma. As for contact lenses loaded with anti-intraocular pressure drugs, drug loading and effective sustained release are major concerns at present. Combining polymer structure and performance endows contact lenses with better performance.

Polymer drug delivery systems face challenges in treating glaucoma, and cross-linking technology or self-assembly strategies offer effective solutions for their design. Cross-linking technology and self-assembly strategies are excellent approaches for enhancing polymer drug delivery systems in glaucoma treatment. Cross-linking generates a robust biological network structure that can efficiently load and release drugs, improving targeting and bioavailability. Additionally, nanoparticles can self-assemble, incorporating various functionalities such as magnetic properties. This self-assembly feature opens up the possibility of precise drug delivery through the use of magnetic fields, which can guide the carrier to the intended target location. This approach has found extensive application in the treatment of cancer-related diseases and offers innovative avenues for addressing glaucoma. Furthermore, the advent of intelligent contact lenses designed for diagnosis and treatment has sparked excitement, as they can monitor IOP and control drug release effectively. This innovation holds great promise for benefiting glaucoma patients. As interdisciplinary fields continue to rapidly develop, numerous innovative polymer drug delivery systems are emerging, presenting opportunities for significant breakthroughs in glaucoma treatment.

## Figures and Tables

**Figure 1 polymers-15-04466-f001:**
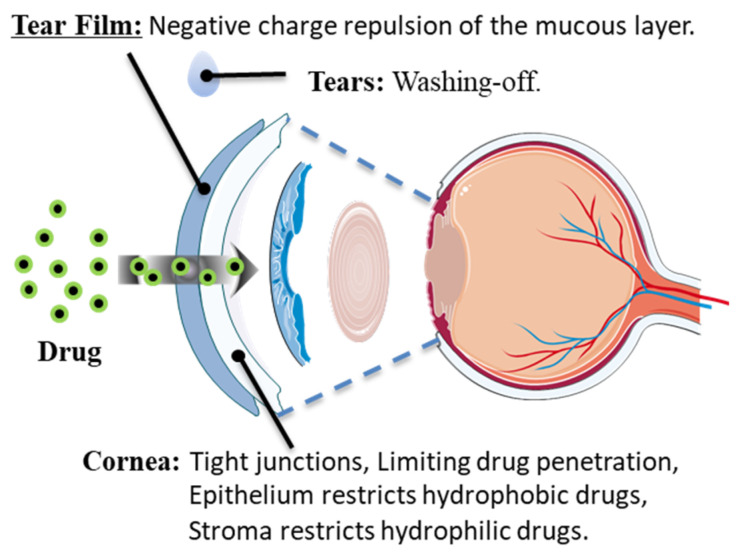
Main barriers to drug absorption on the ocular surface.

**Figure 2 polymers-15-04466-f002:**
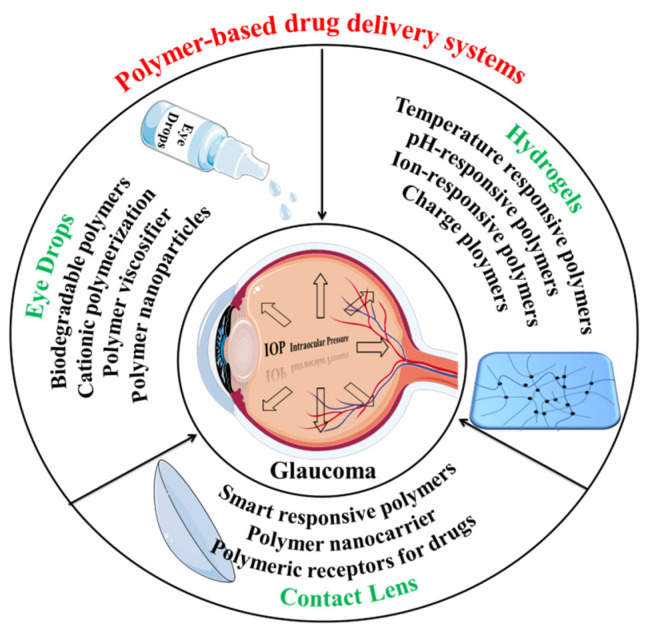
Polymer-based delivery systems for glaucoma treatment: Eye drops, Hydrogels, and Contact lenses (all incorporating polymers).

**Figure 3 polymers-15-04466-f003:**
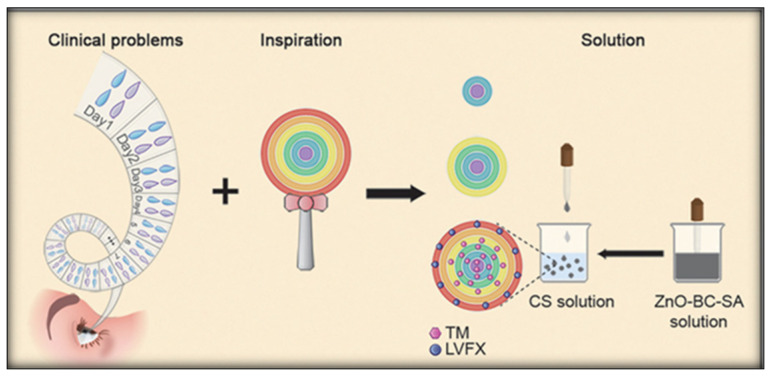
The inspiration for this study: a bull’s-eye-shaped rainbow lollipop. Each layer of the lollipop has a different taste; by placing TM and LVFX in different HB layers, multiple effects and sustained release can be realized. Reproduced from ref. [47] with permission from Wiley-VCH, copyright 2021.

**Figure 4 polymers-15-04466-f004:**
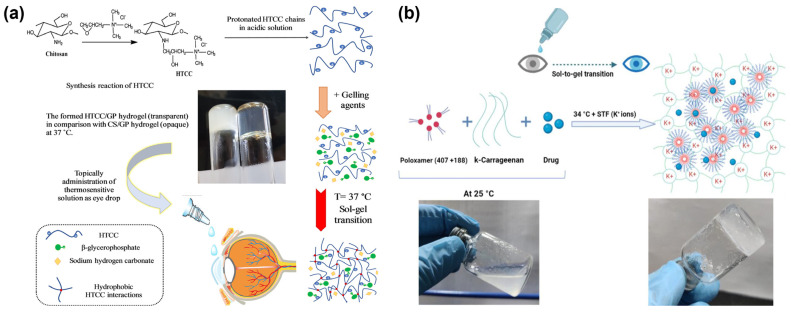
Ocular environmentally responsive gels. (**a**) Schematic of preparation stages for timolol-loaded thermosensitive hydrogel as an ocular drug delivery system for topical administrations. Reproduced from Ref. [65] with permission from Elsevier, copyright 2020. (**b**) Polymer-based delivery systems for illustration, showing the possible gel matrix formed by the NEB-loaded dual-responsive in situ gel at 34 °C in presence of STF (containing K^+^ ions). Reproduced from Ref. [66] with permission from MDPI, copyright 2023.

**Figure 5 polymers-15-04466-f005:**
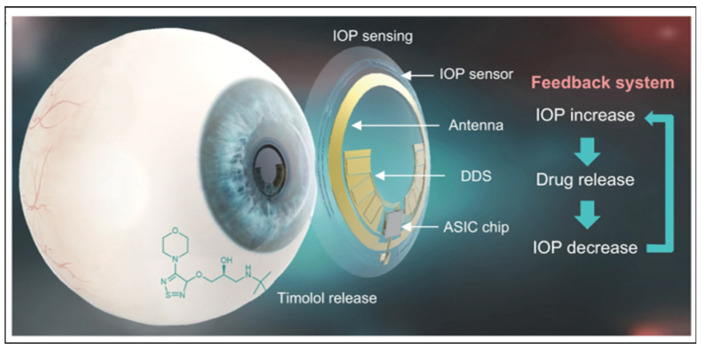
Schematic illustration of a theranostic smart contact lens for glaucoma. Reproduced from Ref. [81] with permission from Springer Nature, copyright 2022.

## Data Availability

Not applicable.

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
