# Peer review of "Polymer-Based Self-Assembled Drug Delivery Systems for Glaucoma Treatment: Design Strategies and Recent Advances"

_polymers, 2023, doi:10.3390/polym15224466_

Round 1
Reviewer 1 Report
Comments and Suggestions for Authors
This manuscript reviewed the strategy of using polymer-based delivery systems for glaucoma treatment including Eye drops, Hydrogels and Contact lenses. The manuscript was well-written. However, some revision is needed.
- The title and introduction focused on self-assembled polymers, but I could see that such use was only in eye drops and hydrogels, but not in contact lenses. Revision is needed to include the use of self-assembled polymers in contact lenses or revise the title and introduction to be only the polymer-based delivery system.
-Section 2.1's title should be changed to Eye Drops since the content also included other techniques for drug delivery improvement, not just self-assembled polymer.
-The reference for the effect of particle size was just only one ([50]). More reviewed references should be included and reviewed.
- Summary and outlook should be revised. At the moment, it was just a repeatedly brief content of the whole manuscript and rather long. This section should not duplicate the content previously described and should focus on the outlook or perspective of the techniques described.
Comments on the Quality of English LanguageSome minor revisions should be conducted for better comprehension.
Author Response
This manuscript reviewed the strategy of using polymer-based delivery systems for glaucoma treatment including Eye drops, Hydrogels and Contact lenses. The manuscript was well-written. However, some revision is needed.
- The title and introduction focused on self-assembled polymers, but I could see that such use was only in eye drops and hydrogels, but not in contact lenses. Revision is needed to include the use of self-assembled polymers in contact lenses or revise the title and introduction to be only the polymer-based delivery system.
Response to the reviewers: Thanks very much for the comment. To better emphasize the role of self-assembling polymers in the preparation of contact lenses, we have comprehensively explained their functions and the process of self-assembly into nanoparticles in the "Contact Lens" section.
Our revision: In the second paragraph of section 2.3, line 2, “Costa et al. reported loading TM into polymer nanoparticles by coupling chitosan with lauric acid and sodium alginate, which were subsequently added to the precursor for contact lens preparation. This resulted in silica gel contact lenses loaded with drugs. After oxygen plasma irradiation and albumin immersion, the lens surface became more hydrophilic and conducive to wear. The positively charged carrier system prolonged drug retention in the cornea and improved bioavailability, with a release duration of up to 6 days. Directly loading drug colloidal particles effectively enhanced drug release performance [72]” has been revised as “Costa et al. reported loading TM into polymer nanoparticles by coupling chitosan with lauric acid and sodium alginate, which were subsequently added to the precursor for contact lens preparation. This resulted in silica gel contact lenses loaded with drugs. The modification of chitosan with lauric acid was crucial to enhance its hydrophobic properties, which proved beneficial for achieving compatibility between the nanostructure and the silicone matrix. The self-assembly and gelation process were facilitated by harnessing the interaction between the positive charge of chitosan and the negative charge on the surface of sodium alginate, ultimately leading to a more stable polymer nanoparticle structure. Subsequent treatment with oxygen plasma irradiation and albumin immersion rendered the lens surface more hydrophilic and conducive to comfortable wear. The positively charged carrier system prolonged drug retention in the cornea and improved bioavailability, with a release duration of up to 6 days. Directly loading drug colloidal particles effectively enhanced drug release performance [73]”
In line 404 in the revised version, “Dang et al. directly loaded polyethylene glycol solid lipid drug nanoparticles into polymer contact lenses, achieving continuous latanoprost release for glaucoma treatment [73]” has been revised as “Dang et al. employed Precirol ATO 5 as the lipid phase, with soy lecithin and Pluronic® F127 serving as surfactants. Additionally, polyethylene glycol monostearate was incorporated into the formulation. They utilized the solvent evaporation method to induce the self-assembly of the components, resulting in the preparation of PEGylated solid lipid nanoparticles loaded with latanoprost. This formulation exhibited the capability to achieve sustained release of latanoprost, making it a promising candidate for the treatment of glaucoma [74].”
We also emphasized the non-covalent interaction applied in the fabrication of a contact len. Lee et al. added TM-loaded thermosensitive poly(N-isopropylacrylamide) nanogel (30~50 nm) to the contact lens via soaking or centrifugation plus soaking. The stability of the self-assembled nanogels was ensured by hydrogen bonding interaction. The lens maintained light transmission and oxygen permeability while exhibiting temperature-triggered sustained drug release at 35℃, providing environmental responsiveness [75].
- Section 2.1's title should be changed to Eye Drops since the content also included other techniques for drug delivery improvement, not just self-assembled polymer.
Response to the reviewers: Thanks very much for the comment. We have already changed the title of Section 2.1 to "Eye drops."
- The reference for the effect of particle size was just only one ([50]). More reviewed references should be included and reviewed.
Response to the reviewers: Thanks very much for the comment. We totally agree with this comment. We have added more references about the effect of particle size. Please find ref50-55 in the revised manuscript.
- Summary and outlook should be revised. At the moment, it was just a repeatedly brief content of the whole manuscript and rather long. This section should not duplicate the content previously described and should focus on the outlook or perspective of the techniques described.
Response to the reviewers: Thanks very much for the comment. The “Summary and Outlook” section has been slimed down. Some unnecessary content has been deleted. Pleas find the “Summary and Outlook” section in the revised version.
- Comments on the Quality of English Language,Some minor revisions should be conducted for better comprehension.
Response to the reviewers: Thanks very much for the comment. We double-checked the manuscript and carefully revised the grammar and spelling errors to improve the quality of English. Please find them in the revised version of the manuscript.

Reviewer 2 Report
Comments and Suggestions for Authors
I would recommend adding a paragraph on the general advantages of polymer-based self-assembled drug delivery systems at the beginning of Section 2, before Section 2.1. This will give the reader a good overview of the potential benefits of these systems before delving into the specific details of the article.
I would also recommend adding a section on the design strategy of penetration into the cornea due to the multiple barriers of the eye.
I would recommend adding a sentence or two about the benefits of the highly branched configuration on line 173.
I would recommend adding a sentence or two about the advantages of thermosensitive hydrogels on lines 213-222.
Are there any clinical stage products using polymer-based self-assembled drug delivery systems for glaucoma treatment?
Regarding line 259-261, explain why low polymer concentration is good, what is the suitable pH range, and why there is no eye irritation?
Is there any biodistribution study on these polymer-based self-assembled drug delivery systems for glaucoma treatment that you can include in this review since you discussed the targeted delivery from line 371 to 377?
Include an example of magnetic nanoparticles in part 2, because these were discuss in the summary and outlook part (Line 406-407)
Comments on the Quality of English LanguageFor the term intraocular pressure, use the abbreviation IOP consistently throughout the article. After the abbreviation is introduced, it should be used in place of the full term "intraocular pressure" unless it is necessary to clarify the meaning.
Author Response
Reviewer 2.
- I would recommend adding a paragraph on the general advantages of polymer-based self-assembled drug delivery systems at the beginning of Section 2, before Section 2.1. This will give the reader a good overview of the potential benefits of these systems before delving into the specific details of the article.
Response to the reviewers: Thanks very much for the comment. We do agree with this comment. Therefore, we added a discussion on the on the general advantages of polymer-based self-assembled drug delivery systems at the beginning of Section 2. Please find it in the 1st paragraph of section 2.
- I would also recommend adding a section on the design strategy of penetration into the cornea due to the multiple barriers of the eye.
Response to the reviewers: Thanks very much for the comment. We have added some discussions on the design strategy of penetration into the cornea. Please find it in the second paragraph of section 2.
- I would recommend adding a sentence or two about the benefits of the highly branched configuration on line 173.
Response to the reviewers: Thanks very much for the comment. We have added some discussions on the benefits of the highly branched configuration. Please find it in the 3rd paragraph of section 2.1.
- I would recommend adding a sentence or two about the advantages of thermosensitive hydrogels on lines 213-222.
Response to the reviewers: Thanks very much for the comment. We have added one or two sentences to discuss on the advantages of thermosensitive hydrogels. Please find them in the second paragraph of section 2.2.
- Are there any clinical stage products using polymer-based self-assembled drug delivery systems for glaucoma treatment?
Response to the reviewers: Thanks very much for the comment. Currently, the primary mode of glaucoma treatment in clinical practice involves the use of eye drops, such as Rhopressa® eye drops by Aerie Pharmaceuticals, designed to reduce intraocular pressure. It is challenging to ascertain whether there are clinical-stage products utilizing polymer-based self-assembling drug delivery systems for glaucoma treatment. This uncertainty arises from the vendors' proprietary information, which typically discloses only the active drug component, leaving the specific delivery system undisclosed. Nevertheless, it is important to note that most recent studies in this field have focused on comparisons with commercially available products. This approach holds the promise of accelerating technological advancements in polymer-based self-assembling drug delivery systems for clinical applications in the treatment of glaucoma.
- Regarding line 259-261, explain why low polymer concentration is good, what is the suitable pH range, and why there is no eye irritation?
Response to the reviewers: Thanks very much for the comment. In the 4th paragraph of Section 2.2, “When compared with thermosensitive and pH-sensitive in-situ gel delivery systems, ion-sensitive in-situ gels offer advantages such as lower polymer concentration, suitable pH, and no eye irritation” has been revised as “When compared with thermosensitive and pH-sensitive in-situ gel delivery systems, ion-sensitive in-situ gels typically require lower polymer concentrations to form a gel due to the involvement of ions. This reduction in polymer concentration helps minimize the amount of polymer needed for ocular applications, lowering the potential risk of adverse reactions. Additionally, ion-responsive hydrogels do not need to be designed with a significant pH difference from the natural pH of tears, which is 7.4. This reduces eye irritation and alleviates patient discomfort.”
- Is there any biodistribution study on these polymer-based self-assembled drug delivery systems for glaucoma treatment that you can include in this review since you discussed the targeted delivery from line 371 to 377?
Response to the reviewers: Thanks very much for the comment. Targeted drug delivery is intricately linked to biodistribution, and in this review, we primarily delve into the ocular surface drug delivery systems designed for the treatment of glaucoma. To provide a comprehensive understanding, we commence by offering a detailed exploration of the metabolic pathways associated with ocular surface drug delivery in the introduction. Throughout this review, we devote significant attention to key factors such as drug residence time within the cornea and the efficacy of drugs, with specific focus on the segments related to eye drops, eye gels, and contact lenses. Our primary emphasis centers on elucidating the intricate process of drug distribution within the anterior segment of the eye. we have included pertinent examples that shed light on ocular drug distribution patterns and metabolic pathways, offering a more complete perspective on this critical aspect of targeted drug delivery for glaucoma treatment. Please find it in 4th paragraph of section 2.1:
“Ali et al. developed a chitosan-based drug delivery system with nanoparticles measuring approximately 243 nm in size. This innovation effectively extended the time the drug remained in the cornea, resulting in a reduction of intraocular pressure in glaucoma patients. However, it's worth noting that after in vivo metabolism and distribution studies, the drug delivery system was swiftly cleared from the corneal region just six hours following intraocular administration. It then proceeded to traverse the nasolacrimal drainage system, ultimately entering the systemic circulation.”
- Include an example of magnetic nanoparticles in part 2, because these were discuss in the summary and outlook part (Line 406-407)
Response to the reviewers: Thanks very much for the comment. Although we have not found the relative literature, we drug delivery through ocular surface is our outlook to the development of the polymer-based self-assembled drug delivery systems for glaucoma treatment. We corrected the relative discussion in the last paragraph of the section 3. Please find it in the revised version of the manuscript.
- Comments on the Quality of English Language For the term intraocular pressure, use the abbreviation IOP consistently throughout the article. After the abbreviation is introduced, it should be used in place of the full term "intraocular pressure" unless it is necessary to clarify the meaning.
Response to the reviewers: Thanks very much for the comment. In the manuscript, the term "intraocular pressure" is mentioned 16 times, excluding the abstract and references sections. To enhance clarity and consistency, we introduced the abbreviation "IOP" after the initial mention in the introduction to denote "intraocular pressure." Consequently, the meaning of "intraocular pressure" was explicitly clarified in two instances, while the remaining 13 instances utilized the abbreviation "IOP" for brevity and consistency.
